# Latent classes associated with the intention to use a symptom checker for self-triage

**Stephanie Aboueid** *, **Samantha B. Meyer, James Wallace, Ashok Chaurasia**

School of Public Health and Health Systems, University of Waterloo, Waterloo, Ontario, Canada

* seaboueid@uwaterloo.ca

**Data Availability Statement:** Data cannot be shared publicly because of the sensitive nature of the data and potential of identifying participants. Data are available from the main researcher team, following ethics approval from the University of

## Abstract

It is currently unknown which attitude-based profiles are associated with symptom checker use for self-triage. We sought to identify, among university students, attitude-based latent classes (population profiles) and the association between latent classes with the future use of symptom checkers for self-triage. Informed by the Technology Acceptance Model and a larger mixed methods study, a cross-sectional survey was developed and administered to students (aged between 18 and 34 years of age) at a University in Ontario. Latent class analysis (LCA) was used to identify attitude-based profiles that exist among the sample while general linear modeling was applied to identify the association between latent classes and future symptom checker use for self-triage. Of the 1,547 students who opened the survey link, 1,365 did not use a symptom checker in the past year and were thus identified as "non-users". After removing missing data (remaining sample = n = 1,305), LCA revealed five attitude-based profiles: *tech acceptors*, *tech rejectors*, *skeptics*, *tech seekers*, and *unsure acceptors*. *Tech acceptors* and *tech rejectors* were the most and least prevalent classes, respectively. As compared to *tech rejectors*, *tech seekers* and *unsure acceptors* were the latent classes with the highest and lowest odds of future symptom checker use, respectively. After controlling for confounders, the effect of latent classes on symptom checker use remains significant (p-value < .0001) with the odds of future use in *tech acceptors* being 5.6 times higher than the odds of future symptom checker use in *tech rejectors* [CI: (3.458, 9.078); p-value < .0001]. Attitudes towards AI and symptom checker functionality result in different population profiles that have different odds of using symptom checkers for self-triage. Identifying a person's or group's membership to a population profile could help in developing and delivering tailored interventions aimed at maximizing use of validated symptom checkers.

## Introduction

Unnecessary care and delaying seeking care are two factors that contribute to higher system costs [1–3]. One way to economize the healthcare system is to provide patients with reliable tools to inform better decisions on when to seek care [1, 4]. Symptom checkers, especially those involving artificial intelligence, have provided a means for users to self-triage (self-assess

Waterloo's Research Ethics Committee (contact: researchoffice@uwaterloo.ca).

**Funding:** The authors received no specific funding for this work.

**Competing interests:** The authors have declared that no competing interests exist.

whether or not they should seek medical care) [5, 6]. Examples of these platforms include Babylon Health, the Ada health app, and the K Health app. Although there are hundreds of symptom checkers available for public use, the literature surrounding the use of this technology remains scarce [7, 8]. It is unclear, for example, whether population groups accept or use this technology as well as the group profiles more likely to accept such a technology.

Research on individual acceptance and use of information technology is one of the most established streams of research in information systems [9]. Stemming from theories in social-psychological and behavioural literature, mainly the Theory of Planned Behavior [10], the Technology Acceptance Model (TAM) outlines various factors to explain an individual's decision to adopt and use a technology [11]. TAM states that behavioural intention, the most proximal santecedent to actual technology use, is influenced by individuals' attitude, which in turn, is influenced by two key constructs: perceived usefulness (PU) and perceived ease of use (PEOU) of the technology [11]. Over time, researchers have applied the TAM to identify factors associated with the use of various types of technologies, in different settings, while targeting diverse population groups. The growing body of knowledge in the field contributed to the development of a refined model, the Unified Theory of Acceptance and Use of Technology (UTAUT) [12].

Most studies applying the TAM and UTAUT frameworks, however, have studied the effect of individual factors on technology use, none of which focused on symptom checkers [8, 12]. For example, higher trust in technology has been shown to be associated with increased technology use, but it is unclear if the co-occurrence of high trust with other attitude-based variables may affect this association. As such it is unclear how a group of variables co-exist and in turn, explain acceptance and use of such symptom checkers. To address this gap, latent Class Analysis (LCA), a statistical and probabilistic method introduced in the 1950s [13], can be used to classify individuals from a heterogeneous group into smaller more homogenous unobserved subgroups [14]. Examples of LCA applications include identifying classes based on Internet searching behaviours among older adults [15], an attitude-based segmentation of mobile phone users [16], and identifying patterns of technology and interactive social media use among adolescents [17]. While there are various possible bases to use in segmentation analysis (e.g., ranging from demographic data to lifestyle-related bases), attitudes have been suggested as a useful basis as they take into account a more affective dimension of consumers' choices and have a better ability to describe behaviour [18, 19].

Little is known about the types of attitude-based population profiles that exist as well as how they are associated with the use of symptom checkers. Addressing this gap has key practical implications for health systems and population health interventions which seek to increased adoption and use of such platforms by the population. The target population in this study were university students as they are typically young adults–a group known to be eager adopters of technology; as such, they are the ideal target for such digital platforms and may contribute to maximizing symptom checker use [20]. The objective of this study was to identify attitude-based latent classes (population profiles) and the association of each of these latent classes with the future use of symptom checkers for self-triage.

## Materials and methods

We conducted a cross-sectional survey-based study that targeted young adults (between the ages of 18 and 34 years of age) enrolled at the university of Waterloo, a public research university with six faculties. Prior to participant recruitment, ethics clearance was granted from the Research Ethics Board at the University of Waterloo (#41366). Participant recruitment occurred through an email invitation sent by the University Registrar's office and a link posted

on the Graduate news webpage. In addition to being approved by the Ethics board, the survey email invitation was also submitted to and approved by the Institute of Analysis and Planning. Consent was obtained from participants through the survey. Data collected cannot be shared for confidentiality purposes.

The survey used in this study (S1 Appendix) was developed and reviewed in collaboration with a Survey Research Center (SRC) at the affiliated University. The SRC is comprised of experts in survey design and methodology who work in developing expertise in rigorous and specialized research. Survey development began in August and was finalized the same year, in December 2020. Survey questions were informed by the literature and adapted for the target population and technology of interest (S2 Appendix). Moreover, to reduce respondent burden, not all factors included in the UTAUT were measured in the survey. A shortlist of factors was developed based on the UTAUT model and a ranking exercise conducted with 22 participants from the same target population (i.e., university students) as part of semi-structured interviews–this list is included in S3 Appendix and findings from this work can be found elsewhere [21].

LCA was used on survey data to identify underlying latent variables based on observed measured categorical variables (i.e., trust, usefulness, credibility, demonstrability, output quality, perspectives about AI, ease of use, and accessibility). The selection of the best fitted latent class model(s) for attitudes towards symptom checker functionality and AI in health was based on key fit statistics and interpretability. For models assessing association between latent classes and future use, our General Linear Logit models considered various types of latent classes, and the best regression model was chosen based on model fits statistics and model interpretability.

### Data set

A total of 35,643 undergraduate university students received an email invitation for the survey through the Registrar's office. A total of 1,547 students complete the survey which was available online on January 11, 2021 and closed the following day. Respondents who clicked on the web survey link and did not complete the survey were classified as either screened out or a drop out. Respondents who were screened out were those not meeting the eligibility criterion of being between the ages of 18 and 34. There were 12 and 2 respondents who indicated they were under 18 or over the age of 34, respectively–they were deemed ineligible and screened out of the survey. Drop-outs were defined as respondents who clicked on the web survey link but did not complete the survey. There was a total of 558 dropouts with just over half (57%) having occurred at the introduction page with the rest of the dropouts occurred throughout the survey with most occurring within the first several questions. Given that the outcome of interest is the future use of symptom checkers, 180 respondents who had used symptom checkers in the past 12 months and were thus categorized as "users" were excluded from the analysis. The remaining sample (n = 1,365) who had not used the platform were identified as "non-users" and are the focus of this study.

### Data analysis

All analyses were performed using SAS 9.4. Descriptive statistics and bivariate analyses were conducted to provide an overview of the sample. Items used to determine latent classes were coded with binary variables such that 1 denoted "no or neutral" and 2 denoted "yes". PROC LCA was used to identify response patterns that define latent classes. In order to identify an optimal baseline model, the procedure was repeated for different numbers of latent classes [22]. Once latent class models were identified, relative model fit statistics were used to select the model that best describes the data. Model selection for best latent class model was based on

goodness of fit measures such as Bayes Information Criterion (BIC) and entropy [23]. A low BIC value, a high entropy value, and interpretability of the classes informed our model selection [22]. General Logit Models were used for our nominal outcome of interest since the three categories do not have a natural order. Future use of symptom checkers was the outcome of interest with it having three categories and "neutral" as the referent categories and the two other categories (i.e., "yes" and "no") compared with this referent. The "neutral" category was used as the referent as the interest was to understand the odds-like of using or not using symptom checkers in the future.

## Results

### Sample

Participants with missing data on key variables of interest were removed (n = 62). The sample (n = 1,305) of non-users is somewhat evenly split across men and women, non-white, enrolled in an undergraduate program, and often have access to the Internet. An overview of this sample in terms of demographics (gender, age, race), academic/professional environment (education level, faculty, employment status), self-perceived health, health literacy, healthcare access, healthcare use, healthcare use frequency, wait time, and healthcare need are shown in Table 1. The counts and percentages of the outcome variable and items used to determine latent classes are presented in Table 2.

### Latent classes

Eight items (i.e., trust, usefulness, credibility, demonstrability, output quality, perspectives about AI, ease of use, and accessibility) were used for latent class modelling; as such, the number of latent class considered were K = 2, 3, . . . 7. Table 3 displays the fit statistics for the LCA for the top three models arising from K = 3,4, and 5 based on fit statistics and interpretability. These models had relatively lower BIC values and higher entropy as shown in Table 3.

Based on the fit statistic and interpretability, the five-class model was chosen. While the BIC and adjusted BIC were slightly higher for the five-class model as compared to the three- and four-class models, the entropy was higher as compared to the 4-class model. Importantly, the five-class model provides more detailed information regarding the classes that exist in the population with *tech seekers* being an important class that is in line with findings from the qualitative phase of this work which highlights the key barrier related to lack of perceived access to symptom checkers. An overview of the five classes are provided in Table 4.

Similarly to the three- and four-latent class models, the first profile describes a group with positive attitudes towards various aspects of symptom checkers and were thusly labeled *tech acceptors*. The second group were the opposite, having a low probability of answering positively on any of the items assessed, and were labeled as *tech rejectors*. The third group had a mixed response pattern showcasing some negative perceptions, particularly related to trust, demonstrability, and output quality–this group was labeled as *skeptics*. The fourth subgroup (*tech seekers*) has positive perceptions related to all aspects of symptom checkers but do not find the platform to be accessible whereas the fifth group (*unsure acceptors*) does not perceive access to be an issue but rather have some negative perceptions about AI and other aspects of symptom checkers.

In terms of prevalence, *tech acceptors* and *tech rejectors* make up the biggest and smallest proportion across models, respectively. *Skeptics* are the second most prevalent group with additional granularity provided in models with additional classes.

**Table 1. Sample characteristics.**

| Characteristics | Count (%) |
|---|---:|
| **Gender** | |
| • Women | 710 (54) |
| • Men | 556 (43) |
| • Other | 39 (3) |
| **Age group** | |
| • 18–24 years | 1256 (96) |
| • 25–29 years | 37 (3) |
| • 30–34 years | 12 (1) |
| **Racial group[a]** | |
| • White | 370 (28) |
| • Non-white | 935 (72) |
| **Current education level[b]** | |
| • Undergraduate | 1272 (97) |
| • Other | 33 (3) |
| **Faculty** | |
| • Engineering | 358 (27) |
| • Sciences | 247 (19) |
| • Applied Health Sciences | 112 (8) |
| • Environment | 77 (7) |
| • Arts | 212 (16) |
| • Mathematics | 299 (23) |
| **Employment status** | |
| • Employed | 469 (36) |
| • Not employed | 785 (60) |
| • Prefer not to disclose | 51 (4) |
| **Self-perceived health[c]** | |
| • Good | 1156 (89) |
| • Poor or do not know | 149 (11) |
| **Health literacy[d]** | |
| • High | 1140 (87) |
| • Average or low | 165 (13) |
| **Healthcare access** | |
| • Same day to 2 weeks | 948 (73) |
| • 2 weeks to 1 month | 85 (7) |
| • One month or more | 24 (2) |
| • Do not know | 248 (19) |
| **Healthcare use[e]** | |
| • Yes | 664 (51) |
| • No or do not know | 641 (49) |
| **Healthcare use frequency[f]** | |
| • None to few | 501 (75) |
| • Sometimes | 120 (18) |
| • Often | 43 (7) |
| **Wait time[g]** | |
| • Short | 982 (75) |
| • Medium or long | 323 (25) |
| **Healthcare need[h]** | |

(*Continued*)

**Table 1.** (Continued)

| Characteristics | Count (%) |
|---|---|
| • Low | 1289 (99) |
| • Medium or high | 16 (1) |

Notes: all percentage values are rounded to the nearest integer.

[a] Race captures the self-perceived racial or cultural group of participants. Prevalent racial groups include South Asian and Chinese. The response options were collapsed into two categories (white and non-white) for data analysis.

[b] Most participants are currently enrolled in an undergraduate program. Masters and PhD programs were grouped into "other".

[c] There were five categories for self-perceived health (i.e., excellent, very good, good, fair, poor) which were grouped into two categories (i.e., good and poor) for data analysis. Eight participants indicated "don't know"; they were grouped with the "poor" self-perceived health group for analysis purposes.

[d] Four questions with five-response option Likert scale were used for measuring health literacy. The mean of the responses was calculated and grouped into three options (i.e., high, average, and low).

[e] Healthcare use was measured by asking whether participants saw a family doctor or nurse in the past year (before COVID-19).

[f] Healthcare use frequency was answered by 664 participants who had utilised healthcare in the past year. Zero to 2 visits were categorized as "none to few"; 3–5 categorized as "sometimes"; and more than 5 visits categorized as "often".

[g] Wait time was measured as the amount of time participants had to wait between the time of their appointment and the time seen by the primary care provider. Less than 15 minutes to 2 hours was categorized as low; 1 to 2 hours was categorized as medium; and 3 hours or more was categorized as long. Eighty-two participants reported long wait times.

[h] Healthcare need was measured by the number of health conditions reported with "no chronic health conditions" and 1–2 health conditions categorized as "low"; 3–5 health conditions categorized as medium; and 6 or more conditions categorized as "high". Four participants were identified to have "high" healthcare need and were grouped with those with medium healthcare need.

## Regression analysis

The GLM procedure in SAS was used to the fit the above General Logit Regression where the five attitude-based latent profiles serve as a predictor variable in regression models. We additionally ran the above models without confounders (i.e., gender, race, healthcare use, wait time, health literacy, and self-perceived health) for the purpose to assess whether the relationship between the main predictor and the outcome changes. Detailed outputs of these model are provided in S5 Appendix. As seen in Tables 5 and 6, it is noteworthy that the effect of latent classes on the future use of symptom checkers remained significant even after controlling for confounders; this highlights the strength of the association between latent classes and symptom checker use.

After controlling for confounders, the effect of latent classes on symptom checker use remains significant (p-value < .0001) with the odds of future use in *tech acceptors* being 5.6 times higher than the odds of future symptom checker use in *tech rejectors* [CI: (3.458, 9.078); p-value < .0001]. The odds of future use are 2.6 times higher in skeptics than the odds of future use in *tech rejectors* [CI: (1.491, 4.586); p-value = .0008]. The odds of future use are 7.6 times higher in *tech seekers* than the odds of future use in *tech rejectors* [CI: (4.276, 13.752); p-value = < .0001]. The odds of future use in *unsure acceptors* are 2 times higher than the odds of future use in *tech rejectors* [CI: (1.207, 3.584); p-value = .008]. In sum, being in a certain latent class is a significant predictor of future symptom checker use. *Tech seekers* and *unsure acceptors* were the latent classes with the highest and lowest odds of future symptom checker use, respectively.

**Table 2. Descriptive statistics on the intent to use symptom checkers.**

| Characteristics | Count (%) |
|---|---:|
| **Future SC use (outcome variable)** | |
| • No | 215 (16) |
| • Neutral | 391 (30) |
| • Yes | 699 (54) |
| **Perspective on the use of AI** | |
| • Negative or neutral | 480 (37) |
| • Positive | 825 (63) |
| **Perceived SC ease of use** | |
| • Low or neutral | 469 (36) |
| • Yes | 836 (64) |
| **Perceived access to SC** | |
| • Low or neutral | 397 (30) |
| • High | 908 (70) |
| **Demonstrability** | |
| • Low or neutral | 644 (49) |
| • High | 661 (51) |
| **Trust** | |
| • Low or neutral | 827 (63) |
| • High | 478 (37) |
| **Usefulness** | |
| • Low or neutral | 318 (24) |
| • High | 987 (76) |
| **Output quality** | |
| • Low or neutral | 442 (34) |
| • High | 863 (66) |
| **Credibility** | |
| • Low or neutral | 161 (12) |
| • High | 1144 (88) |

Notes: all percentage values are rounded to the nearest integer; variables in the table were measured using Likert scale response options.

## Discussion

To our knowledge, our study is the first to merge the TAM and LCA literature to identify profiles among university students and regress these profiles on future symptom checker use. Interestingly, while young adults are perceived to be technology savvy, most of the participants recruited had not used a symptom checker in the past year–this may be due to the lack of awareness regarding the existence of these platforms [21]. Most had positive perspectives regarding the use of AI in health and symptom checkers' functionality; however, some skepticism and issues related to perceived accessibility and functionality may hinder the future adoption and use of symptom checkers. Five distinct latent classes were identified: *tech acceptors*, *tech rejectors*, *skeptics*, *unsure acceptors*, and *tech seekers*. It is a noteworthy finding that the effect of latent classes remained significant even after controlling for confounders; this is not always the case since from a statistical perspective, the effect of a variable can lose its significance when controlling for other variables [24].

Previous studies have applied the TAM to identify the factors associated with the adoption and use of health apps and health technologies; for example, a study found that adolescents

**Table 3. Fit statistics for the latent class analysis.**

|  | Number of latent classes | | | | | |
|---|---|---|---|---|---|---|
|  | **2** | **3** | **4** | **5** | **6** | **7** |
| **Fixed effects model** | | | | | | |
| Degrees of freedom | 238 | **229** | **220** | **211** | 202 | 193 |
| Log likelihood | -5882.62 | **-5837.06** | **-5802.22** | **-5786.55** | -5776.10 | -5768.13 |
| G-squared | 392.99 | **301.87** | **232.19** | **200.85** | 179.96 | 164.01 |
| AIC | 426.99 | **353.87** | **302.19** | **288.85** | 285.96 | 288.01 |
| BIC | 514.95 | **488.40** | **483.28** | **516.51** | 560.18 | 608.80 |
| Adjusted BIC | 460.95 | **405.81** | **372.10** | **376.74** | 391.83 | 411.85 |
| Entropy | 0.74 | **0.65** | **0.61** | **0.63** | 0.63 | 0.66 |

Note: The bolded text represents models (3, 4, and 5 latent classes) that have been interpreted further for their potential in being selected as the preferred model. An interpretation of these models are in a S4 Appendix.

found wearable activity trackers to be useful, but the efforts required to use these technologies may influence overall engagement and technology acceptance [25]. In our study perceived ease of use was also found to play a role in defining latent classes and in turn, the latent class association with future use of symptom checkers. For example, *tech rejectors* and *unsure acceptors* did not perceive the use of symptom checkers to be easy which was evident by their lower odds of using symptom checkers in the future. While age was not explored in our study due to the young age of our sample, another study found that younger populations displayed more confidence with the use of mHealth apps and were less concerned about compromising the confidentiality of their health records [26]. Answers to TAM-related questions among mHealth apps users were significantly more positive compared with non-users [26]. Interestingly, as found in our study, the endorsement of health apps by health organizations can play an influential role in technology acceptance and utilization as well as support efforts in shaping regulation [26, 27].

Tech seekers and unsure acceptors had the highest and lowest odds of future symptom checker use, respectively. Interestingly, it was found that *tech seekers* (those who have positive perspectives related to symptom checker functionality and AI but do not perceive to have access to the technology) had the highest odds of future symptom checker use, even more so than *tech acceptors* (those who have positive perspectives related to all aspects and perceive to have access to the technology). This nuance was highlighted through five latent classes but lost

**Table 4. Five-latent-class model: Probability of positive perceptions for each subgroup.**

|  | Latent Class (count; %) | | | | |
|---|---|---|---|---|---|
|  | Tech acceptors (621, 48%) | Tech rejectors (137, 11%) | Skeptics (190, 14%) | Unsure acceptors (185, 14%) | Tech seekers (172, 13%) |
| Trust | **0.5428** | 0.0675 | 0.1217 | 0.1887 | **0.5521** |
| Credibility | **0.9927** | 0.3112 | **0.7544** | **0.9744** | **0.9724** |
| Output quality | **0.8824** | 0.0924 | 0.3572 | **0.5811** | **0.8679** |
| Usefulness | **0.9671** | 0.0989 | **0.5600** | **0.7480** | **0.8479** |
| Demonstrability | **0.7195** | 0.1102 | 0.2649 | 0.1678 | **0.8359** |
| Accessibility | **0.9939** | 0.1905 | **0.8921** | **0.5369** | 0.1311 |
| Ease of use | **0.8036** | 0.2076 | **0.8729** | 0.3697 | **0.5082** |
| Perspectives about AI | **0.7557** | 0.3517 | **0.5774** | 0.4656 | **0.7249** |

Note: Item-response probabilities >.5 are bolded to facilitate interpretation.

**Table 5. Output for the five-class model without confounders.**

| Type 3 Analysis of Effects | | | |
|---|---|---|---|
| **Effect** | **DF** | **Wald Chi-Square** | **Pr > ChiSq** |
| Latent Class | 8 | 142.8164 | < .0001 |

when approaching the same objective with three or four latent classes. These classes could serve as a starting point in similar studies targeting other population groups.

This study has several strengths that relate to the technology studied, choice of target population, theoretical framework and methodological approach used, tools developed, and practical implications for key stakeholders in the public health arena. Firstly, the development and use of an interview protocol and survey will enable other researchers in the field to adapt and use these tools. This study also contributed to developing the literature on an understudied technology that has real potential in addressing key healthcare challenges. Symptom checkers, along with other digital platforms that allow for self-care, have been named as one of the top 10 emerging technologies in 2020 [28], and their importance has been accentuated during the COVID-19 pandemic [29]. Our study allowed for the identification of five latent classes that may need to be targeted differently to promote the use of promising symptom checkers.

Some limitations warrant mention. First, findings stem from a bounded case which is categorized by a sample that is highly educated and perceived to have a good health status thus limiting the transferability of findings to other populations with a wide range of age groups, education levels, self-perceived health, and health literacy. As such, additional studies targeting other population groups are needed. Moreover, selection bias may be present as those included

**Table 6. Output for the five-class model with confounders.**

| Type 3 Analysis of Effects | | | |
|---|---|---|---|
| **Effect** | **DF** | **Wald Chi-Square** | **Pr > ChiSq** |
| Latent Class | 8 | 143.3710 | < .0001 |
| GenHealth[1] | 2 | 2.7162 | 0.2572 |
| HL[2] | 2 | 0.6488 | 0.7230 |
| HC Use[3] | 2 | 5.6047 | 0.0607 |
| Wait time[4] | 2 | 5.0084 | 0.0817 |
| Gender[5] | 4 | 5.8547 | 0.2103 |
| Race[6] | 2 | 12.3150 | 0.0021 |

| Odds Ratio Estimates | | | | |
|---|---|---|---|---|
| Effect | Future Use | Point Estimate | 95% Wald Confidence Limits | |
| *Tech acceptors* vs. *tech rejectors* | Yes | 5.603 | 3.458 | 9.078 |
| *Tech acceptors* vs. *tech rejectors* | No | 0.565 | 0.346 | 0.922 |
| *Skeptics* vs. *tech rejectors* | Yes | 2.615 | 1.491 | 4.586 |
| *Skeptics* vs. *tech rejectors* | No | 1.384 | 0.808 | 2.371 |
| *Tech seekers* vs. *tech rejectors* | Yes | 7.669 | 4.276 | 13.752 |
| *Tech seekers* vs. *tech rejectors* | No | 0.662 | 0.325 | 1.352 |
| *Unsure acceptors* vs. *tech rejectors* | Yes | 2.080 | 1.207 | 3.584 |
| *Unsure acceptors* vs. *tech rejectors* | No | 0.538 | 0.302 | 0.958 |

[1] Self-perceived health

[2] Health literacy

[3] Healthcare use.

in the study may be different than those who did not opt to participate; however, findings from this work could help reduce selection bias in future studies as it provides an overview of the profiles that may exist and thus, should be represented in the sample. While the study targeted adults between the ages of 18 and 34, most participants were between 18 and 24 suggesting that latent classes identified may differ if the sample was comprised of individuals in the higher age range. This study focused specifically on non-users with the intention to use a symptom checker being the outcome of interest; while data on "users" were collected, the sample size was too small highlighting that a higher sample size will be required to avoid underextraction of classes. Survey questions were not assessed for two psychometric measures (i.e., reliability and validity); however, questions were developed based on published studies and adapted for the target population and technology. Moreover, the survey was developed with assistance from the Survey Research Center; as such, best available practices were applied in survey design, administration, collection and curation.

## Conclusion

Symptom checkers may not be as widely known by the population, even those considered to be eager adopters of technology. Within the university student population, profiles–characterized by their attitudes toward symptom checkers and AI–exist. Perceived ease of use and accessibility are key factors that explain some of the nuances across identified profiles. To maximize the use of validated symptom checkers and therefore, reduce unnecessary healthcare visits, targeted interventions could be developed and delivered depending on an individual's or group's identification to a certain profile. Future research is warranted to assess whether similar profiles exist among other population groups as well as which interventions (both at the health system and population health levels) would be best suited based on existing attitude-based variables.

## Supporting information

**S1 Appendix. Survey.**
(DOCX)

**S2 Appendix. Construct definitions and source of survey questions.**
(DOCX)

**S3 Appendix. Number of participants choosing factors that are important for using a symptom checker for self-triage.**
(DOCX)

**S4 Appendix. Interpretation of the three- and four-latent class models.**
(DOCX)

**S5 Appendix. Detailed GLM output.**
(DOCX)

## Acknowledgments

The authors would like to thank the university students who agreed to participate in the study as well as the administration staff at the University of Waterloo for aiding with participant recruitment.

## Author Contributions

**Conceptualization:** Stephanie Aboueid.

**Formal analysis:** Stephanie Aboueid, Ashok Chaurasia.

**Investigation:** Stephanie Aboueid, Samantha B. Meyer, Ashok Chaurasia.

**Methodology:** Stephanie Aboueid, Samantha B. Meyer, James Wallace, Ashok Chaurasia.

**Project administration:** Stephanie Aboueid.

**Resources:** Stephanie Aboueid, Samantha B. Meyer, James Wallace, Ashok Chaurasia.

**Software:** Ashok Chaurasia.

**Supervision:** Samantha B. Meyer, James Wallace, Ashok Chaurasia.

**Validation:** Stephanie Aboueid.

**Writing – original draft:** Stephanie Aboueid.

**Writing – review & editing:** Samantha B. Meyer, James Wallace, Ashok Chaurasia.

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
