## [Decision Letter · Decision Letter 0]

18 Aug 2021

PONE-D-21-24082

Latent Classes Associated with the Future Use of a Symptom Checker for Self-Triage by Young Adults

PLOS ONE

Dear Dr. Aboueid,

Thank you for submitting your manuscript to PLOS ONE. After careful consideration, we feel that it has merit but does not fully meet PLOS ONE’s publication criteria as it currently stands. Therefore, we invite you to submit a revised version of the manuscript that addresses the points raised during the review process.

ACADEMIC EDITOR: Please see comments below

We look forward to receiving your revised manuscript.

Kind regards,

Dejan Dragan, PhD

Academic Editor

PLOS ONE

Journal Requirements:

2. In the Methods section of the manuscript, please provide additional information regarding the steps taken to validate the questionnaire used in the study.

3. We note that the figure in your supporting information contain copyrighted images. All PLOS content is published under the Creative Commons Attribution License (CC BY 4.0), which means that the manuscript, images, and Supporting Information files will be freely available online, and any third party is permitted to access, download, copy, distribute, and use these materials in any way, even commercially, with proper attribution. For more information, see our copyright guidelines: http://journals.plos.org/plosone/s/licenses-and-copyright.

a. You may seek permission from the original copyright holder of Figure to publish the content specifically under the CC BY 4.0 license. 

Additional Editor Comments:

The reviewers have completed their reviews. Three of them require a minor revision, while the fourth insists on the major revision. My decision is: A major revision. Please, follow all comments carefully and fix them. AE DD

Reviewers' comments:

Reviewer's Responses to Questions

**Comments to the Author**

1. Is the manuscript technically sound, and do the data support the conclusions?

Reviewer #1: Yes

Reviewer #2: Yes

Reviewer #3: Yes

Reviewer #4: Yes

2. Has the statistical analysis been performed appropriately and rigorously? 

Reviewer #1: Yes

Reviewer #2: Yes

Reviewer #3: Yes

Reviewer #4: Yes

3. Have the authors made all data underlying the findings in their manuscript fully available?

Reviewer #1: Yes

Reviewer #2: No

Reviewer #3: No

Reviewer #4: No

4. Is the manuscript presented in an intelligible fashion and written in standard English?

Reviewer #1: Yes

Reviewer #2: Yes

Reviewer #3: Yes

Reviewer #4: Yes

5. Review Comments to the Author

Reviewer #1: This manuscript analyzed the hidden factors for future use of a AI-based symptom checker for self-triage. Though this manuscript is well-written, I have sever comments.

Major comments:

1) Why the authors focus on young adults? Is there any specific reason without comparison on older adults?

2) Why the authors choose only "non-users" for their analysis? It should be comparison results between users and non-users.

3) Why five latent classes are chosen for future analysis? It seems too arbitrary.

4) How the authors control the confounders? There is no explanation on controlling confounders. Please describe it.

5) What are the exact number of samples? It is confusing.

A total of 1,547 students. 12 (under 18) and 2 (over 34) are removed. This means 1533 (1547 - 14). And 180 respondents are users. This derives 1533-180=1353.

But the manuscript mentioned 1,365 samples and after removing missing data, 1,305. Which one is correct?

Minor comments:

6) The legends of Table 1 and Table 2 are the same. Please revise it.

Reviewer #2: In their manuscript, Aboueid et al. present results from a survey among university students aged 18-34 years regarding their attitude towards, and use of, symptom checkers used for self-triage. They identify 5 latent classes representing attitude profiles - tech acceptors, tech rejectors, skeptics, tech seekers, and unsure acceptors. According to their analysis, tech seekers are the most likely and tech rejectors are the least likely to use symptom checkers in the future. While their results provide interesting and novel insights into attitudes that may promote or hinder the acceptance of symptom checkers in different population subgroups, several shortcomings need to be addressed before publication of this manuscript.

1. The study likely suffers from self-selection bias because it only includes participants who opened a link in an email. This should at least be discussed.

2. How were the items on the questionnaire chosen?

3. The latent classes are inferred from “non-users”, and the authors give transparent reasons for this choice. However, they should retrospectively determine in which latent class the “users” fall - are they e.g. tech acceptors or seekers?

4. The fit statistics are not substantially worse for the cases of fewer latent classes. If a smaller number of latent classes (2, 3 or 4) is chosen, do the 5 classes end up in separate classes? This would validate their separation; it is hinted at in the manuscript but should be demonstrated explicitly.

5. A major shortcoming of this study is that it was only conducted in one defined population. It would substantially increase the impact of the study if the authors could confirm whether equivalent latent classes are also observed in other populations. At the very least, this shortcoming should be discussed in the manuscript.

6. Why do tech seekers have higher odds of symptom checker use than tech acceptors, despite finding them less accessible? This is counter-intuitive and should be discussed in detail.

7. The authors mention the significance of their work for public health, but the immediate impact of their results is not clear. They argue that members of the different latent classes may need to be targeted differently in public health interventions, but apart from ease of use or access to symptom checkers, their results suggest that a more general lack of trust in AI is also shown by some participants. It would be very helpful if the authors could expand on this point, suggesting explicitly in what way their results could be taken into account for public health interventions.

Reviewer #3: The manuscript provides an interesting insight into the attitudes of unversity students towards symptom checker applications. It is well written and even more details are presented in the supplement. However, the raw data are not published.

I have some minor questions/comments to the manuscript:

1. Page 5, row 115: the participants of the interviews were from the same population as the study targets, i.e. university students? please indicate this in the text

2. Page 9, row 173: why is "gender" mentioned in the footnote for race? please review and correct if necessary

3. Page 9, row 178: "fair" was also categorized into "good health", and only "poor" (and "don't know") were grouped into the main category "poor"?

4. Page 13, Table 6: footnotes 4-6 only repeat the content of the table cell, I think they are not necessary and can be removed

Reviewer #4: I overall enjoyed reading the work, also, due to its high relevance and real life applicability. I only have some minor questions about details.

Introduction:

- In the introduction the purpose, the necessity and the real life application of the study is clearly stated and explained

- It is also clear which models are used and why these models are chosen

Materials and methods:

- Overall this chapter is also described really clearly

- There is only one thing, which I do not really understand. How are the eight categorical variables chosen? In the bottom of page fife is written: “LCA was used on survey data to identify underlying latent variables based on observed measured categorical variables (…).”

Please specify the process, in which the categorical variables were observed and measured

Data set:

The data set is very clearly and precisely described.

Data Analysis:

The data analysis part is clearly and precisely described.

Results (Sample):

- Tabel1:

- Even though the purpose of table 1 is clear, it is not part of the main result and rather a clarification of the sample group. Thus, it is part of the supportive material

- There are some highly unbalanced classes in the table. In some cases, you could even argue to remove the underrepresented class and give a statement about a narrower sample group (for example a separation into 1256, 37 and 12 (age group) data points is problematic, since the outcome basically represents an age group of 18-24). Since the goal of the study is to find latent classes and not comparing the demographic groups, one option is to keep the data as it is. But then a disclaimer in the discussion mentioning the distribution / group imbalance is necessary to outline this issue and the limitation to the study

- Table2:

- As already mentioned in materials and methods I do not understand where the eight parameters are based on. If I compare the parameters from table 2 to the supporting material 2, I can identify the following:

o Trust

o Credibility

o Perceived accessibility

o tangibility of the result(s) = Demonstrability ?

o healthcare need = Usefulness ?

o Perspective on the use of AI (from TAM ? )

o Perceived SC ease of use (from TAM ? )

o Output quality (from TAM ? )

- Please clarify in chapter materials and methods or in results

- Table 1 and table 2 have both the same title. Please use titles which supply supporting information about the data show in the table

Results (Latent Classes):

- In the discussion is written why the 5 class model is better interpretable than the 3 or 4 class model. Please enter a similar explanation here. Otherwise, it seems like an unsubstantial comment

- Sentence: “While the BIC and adjusted BIC were slightly higher for the five-class model as compared to the three- and four-class models.”

If I compare that statement to table 3 it is not entirely correct. Adjusted BIC is for the 3 class model higher than for the 5 class model.

Results (Regression Analysis):

Is well explained

Discussion:

- Overall it is a good discussion outlining the strengths and limitations of the study

- Please add the disclaimer about the sample group distribution

- there are two "most" in the discussion. Please add specific numbers

6. PLOS authors have the option to publish the peer review history of their article (what does this mean?). If published, this will include your full peer review and any attached files.

Reviewer #1: No

Reviewer #2: No

Reviewer #3: **Yes: **Tamas Toth

Reviewer #4: No

---

## [Author Response · Author response to Decision Letter 0]

23 Sep 2021

RESPONSE TO REVIEWERS 

Dear editorial board and reviewers, 

We first want to thank you for the time you have taken to review our work. 

We sought to address journal requirements. Below is a record of responses from the previous revision. 

Previous submission: 

We are responding to your comments regarding the manuscript entitled “Latent Classes Associated with the Intention to Use a Symptom Checker for Self-Triage”. We hope that the changes and clarifications made to the manuscript address your comments. 

Journal requirements 

• We have adapted the manuscript to adhere to journal requirements. 

Method for validating the questionnaire 

• We added supporting information to highlight the definition of constructs measured as well as the sources of questions used. These questions were adapted for the target population and technology of interest in collaboration with the Survey Research Center at the University of Waterloo. This is also now explained in the methods section. 

Figure in the survey 

• This figure has been removed and replaced with a description of the image. 

Data availability 

• Three reviewers have noted that data were not made publicly available. For confidentiality purposes, these data cannot be shared. As part of the data collection process and ethics clearance, participants were told that data will not be shared or used for purposes other than for this study. As per PlosOne data privacy instructions, this has been indicated in the “materials and methods” section. 

Reviewer 1: 

Major comments: 

1. Why the authors focus on young adults? Is there any specific reason without comparison on older adults? 

To limit the scope of work, we focused on young adults because they are typically eager adopters of technology and tend to be technology savvy. As such, they could be considered the “ideal target” for such platforms. If this population has negative attitudes towards symptom checkers, then it may be likely that older adults (typically less eager to adopt new technologies) will also have this attitude. Moreover, young adults are undergoing a transition period in which they have to start making decisions regarding their own health and more likely to engage in risky behaviours. It was an interest for myself and my co-authors to explore how symptom checkers could help address health needs of this population. 

We acknowledge the importance of conducting other similar studies in other age groups. The profiles identified in this work could inform future work with older adults and would thus help with replication. 

2. Why the authors choose only "non-users" for their analysis? It should be comparison results between users and non-users. 

The focus of this study was to identify profiles within a population rather than compare two populations. The survey collected information on both “users” and “non-users”; however, the “users” sample was only 180 which was not sufficient for us to conduct LCA (Dziak et al., 2014). While the researchers initially thought that symptom checkers was widely adopted by university students, both our qualitative and quantitative work found that symptom checkers were not as known. 

3. Why five latent classes are chosen for future analysis? It seems too arbitrary. 

Given that we have eight items, the minimum and maximum number of classes is 2 and 7 (respectively), within which lies the most optimal number of classes. The choice of the optimal number of classes was based on three criteria: model fit statistics, nature of the determined classes (based on the 8 items), and their interpretability. Once the nature of latent classes was determined for each of the number of classes (from 2 to 7), the optimal model was selected based in the aforementioned criteria. This process of model selection in LCA is guided/recommended by the original authors of LCA methodology Lanza’s (2007). Hence, the model selection for best latent class model was not arbitrary. Additionally, the choice of the five class model elucidated a fifth distinct class (in terms of the behaviour profile in the 8 items), which is the sole purpose of LCA methodology – determine distinct classes/patterns formed by the responses in the eight items. While the BIC and adjusted BIC were slightly higher for the five-class model as compared to the three- and four-class models, the entropy was higher as compared to the 4-class model. More importantly, the five-class model provides insight into existence of the (new) class/profile described as tech seekers. This finding is in line with findings from the qualitative phase (Aboueid et al., 2021) which highlighted perceived access to symptom checkers as one of the key barriers. Models with 6 or 7 classes did not provide optimal fit statistics or meaningful class descriptors/interpretations and hence did not serve as top candidate models for the data. 

4. How the authors control the confounders? There is no explanation on controlling confounders. Please describe it. 

Variables that were used as confounders in our study were based on literature pertaining to technology acceptance model. Our model(s) controlled for the confounding effects of (i) health literacy, (ii) perceived health, and other variables highlighted in the supporting document. The key takeaway from these models is that a subject’s latent class is significant predictor of intention of future even after controlling for effects of confounders the influence intention of future use. This finding elucidates that profile of subjects based on their attitudes towards SC, is an important factor when determining intention of future use of SC, even after controlling for subject’s confounding effects. 

5. What are the exact number of samples? It is confusing. 

A total of 1,547 students. 12 (under 18) and 2 (over 34) are removed. This means 1533 (1547 - 14). And 180 respondents are users. This derives 1533-180=1353. 

But the manuscript mentioned 1,365 samples and after removing missing data, 1,305. Which one is correct? 

The 1,547 does not include those that were screened out and drop-outs. The 1,547 are the number of surveys that were completed. So the calculation is as such: 1,547 – 180 = 1,367 (non-users). 

1,367 – 62 (participants with missing data) = 1,305. The numbers have been adjusted to reflect this. 

Minor comments: 

6. Thank you – this has been revised. 

Reviewer 2: 

1. The study likely suffers from self-selection bias because it only includes participants who opened a link in an email. This should at least be discussed. 

We agree about this point and we have made this more explicit in the limitations. 

2. How were the items on the questionnaire chosen? 

This study was part of a larger mixed methods study in which the first phase of the work entailed conducting interviews with participants of the same target population. Based on our literature review of the technology acceptance model, there is a total of over 15 variables/factors that could potentially be important. To reduce respondent burden, 22 participants were asked to choose the top five factors from the 15 potential factors they believed to be most important when deciding to use symptom checkers. Factors the at were chosen most often in addition the factors identified by Davies et al. were then selected as the varaibles to be measured in the survey. This information is now incorportated in the methods section. 

3. The latent classes are inferred from “non-users”, and the authors give transparent reasons for this choice. However, they should retrospectively determine in which latent class the “users” fall - are they e.g. tech acceptors or seekers? 

Thank you for this feedback. The issue with predicting classes for “users” based on model built using “non-users” assumes subject are exchangeable between the two groups, which is not an assumption we feel comfortable making for our study. The two groups may come from different populations and extrapolation from one group to another can be misleading. We aimed to conduct the analysis for the “users” but the sample is too small (in some cases the cell size was less than 5 participants in certain matrices); as such, the findings would not be reliable. Additional participants will be required for future studies to assess the profiles among those users as well as their association with use and frequency of use. 

4. The fit statistics are not substantially worse for the cases of fewer latent classes. If a smaller number of latent classes (2, 3 or 4) is chosen, do the 5 classes end up in separate classes? This would validate their separation; it is hinted at in the manuscript but should be demonstrated explicitly. 

We agree that the fit statistics for the 3 and 4 classes are not much worse, but the five-class model highlighted an interesting split of the unaware acceptors which showed that “perceived accessibility” to be a key hindrance which is similar to what was found in our qualitative work. Interpretability was the key factor that guided the choice of the 5-class model. 

5. A major shortcoming of this study is that it was only conducted in one defined population. It would substantially increase the impact of the study if the authors could confirm whether equivalent latent classes are also observed in other populations. At the very least, this shortcoming should be discussed in the manuscript. 

We have addressed this point in length in the first comment made by reviewer 1. We are hoping that this study and approach will be leveraged by other researchers in the field who will target other population groups. This will be important for replication and generalizability. We have made this point more explicit in our limitations. 

6. Why do tech seekers have higher odds of symptom checker use than tech acceptors, despite finding them less accessible? This is counter-intuitive and should be discussed in detail. 

We too found this quite interesting as well but are unsure why this is the case. The reason behind such an association eludes and hence, rather the speculating reasons for such a counter-intuitive finding, our discussion leaves this as an association to be investigated in future studies with different populations. 

7. The authors mention the significance of their work for public health, but the immediate impact of their results is not clear. They argue that members of the different latent classes may need to be targeted differently in public health interventions, but apart from ease of use or access to symptom checkers, their results suggest that a more general lack of trust in AI is also shown by some participants. It would be very helpful if the authors could expand on this point, suggesting explicitly in what way their results could be taken into account for public health interventions. 

The items measured in this survey could also be assessed in different settings depending on data captured (e.g., in a healthcare clinic or a national survey). It would then be possible to tailor discussions and interventions based on the profile of a person or group. Trust was perceived to be quite important throughout our work, so interventions and strategies related to improving trust in symptom checkers is important; however, if a person does trust the symptom checker but does not perceive it to be accessible or easy to use, other interventions may need to be used such as referring the patient to individuals who could support in symptom checker use etc. Identifying interventions and strategies for this was not in the scope of this work but is an area of exploration for us. 

Reviewer 3: 

1. Page 5, row 115: the participants of the interviews were from the same population as the study targets, i.e. university students? please indicate this in the text 

This has been clarified. 

2. Page 9, row 173: why is "gender" mentioned in the footnote for race? please review and correct if necessary 

Thanks for spotting that – it is now corrected. 

3. Page 9, row 178: "fair" was also categorized into "good health", and only "poor" (and "don't know") were grouped into the main category "poor"? 

The interest was to compare to those who had poor health and the variable had to be dichotomized for analysis purposes. As such, any category that was not “poor” or “don’t know” were not included in the “good” category. Given that the “good” category includes those who have excellent to fair health, the sample averages out and thus justify the label “good”. Moreover, even if the “fair” group was added into the “poor” self-perceived health category, the analysis would not change. 

4. Page 13, Table 6: footnotes 4-6 only repeat the content of the table cell, I think they are not necessary and can be removed 

Agreed. They have been removed. 

Reviewer 4: 

Introduction: 

1) In the introduction the purpose, the necessity and the real-life application of the study is clearly stated and explained 

Thank you. 

2) It is also clear which models are used and why these models are chosen 

Thank you. 

Materials and methods: 

3) Overall this chapter is also described really clearly 

Thank you. 

4) There is only one thing, which I do not really understand. How are the eight categorical variables chosen? In the bottom of page fife is written: “LCA was used on survey data to identify underlying latent variables based on observed measured categorical variables (…).” 

Please specify the process, in which the categorical variables were observed and measured 

The eight variables were chosen based on input from participants in the qualitative phase on this work. Prior to developing the survey, 22 participants from the same target population were asked to choose the top five factors they believed to be most important when deciding to use a symptom checker for self-triage. The ones most chosen were the ones included in the survey in addition to the factors identified as important by Davies (i.e., perceived ease of use and perceived usefulness). 

Data set: 

5) The data set is very clearly and precisely described. 

Thank you. 

Data Analysis: 

6) The data analysis part is clearly and precisely described. 

Thank you. 

Results (Sample): 

- Tabel1: 

7) Even though the purpose of table 1 is clear, it is not part of the main result and rather a clarification of the sample group. Thus, it is part of the supportive material 

Our reason for keeping table 1 in the main text is to gives the reader some context about the characteristics of the sample. This will allow them to read the rest of the results with sample characteristics in mind. For this reason, the revised manuscript has table 1 in the main text instead of supplementary/supportive material. 

8) There are some highly unbalanced classes in the table. In some cases, you could even argue to remove the underrepresented class and give a statement about a narrower sample group (for example a separation into 1256, 37 and 12 (age group) data points is problematic, since the outcome basically represents an age group of 18-24). Since the goal of the study is to find latent classes and not comparing the demographic groups, one option is to keep the data as it is. But then a disclaimer in the discussion mentioning the distribution / group imbalance is necessary to outline this issue and the limitation to the study 

We have incorporated the your comments in our limitation section. 

- Table2: 

9) As already mentioned in materials and methods I do not understand where the eight parameters are based on. If I compare the parameters from table 2 to the supporting material 2, I can identify the following: 

o Trust 

o Credibility 

o Perceived accessibility 

o tangibility of the result(s) = Demonstrability ? 

o healthcare need = Usefulness ? 

o Perspective on the use of AI (from TAM ? ) 

o Perceived SC ease of use (from TAM ? ) 

o Output quality (from TAM ? ) 

- Please clarify in chapter materials and methods or in results 

Thank you for this feedback. This has been addressed in our responses for Reviewer 1. 

10) Table 1 and table 2 have both the same title. Please use titles which supply supporting information about the data show in the table 

Thanks for spotting this – we edited the titles accordingly. 

Results (Latent Classes): 

11) In the discussion is written why the 5-class model is better interpretable than the 3 or 4 class model. Please enter a similar explanation here. Otherwise, it seems like an unsubstantial comment 

We have added this explanation in the results. 

12) Sentence: “While the BIC and adjusted BIC were slightly higher for the five-class model as compared to the three- and four-class models.” 

If I compare that statement to table 3 it is not entirely correct. Adjusted BIC is for the 3 class model higher than for the 5 class model. 

We have corrected this. Interpretability was the main reason why we opted for the five-class model rather than the two other candidates (3- and 4- class models). Please see our response in comment #3 of Reviewer1. 

Results (Regression Analysis): 

13) Is well explained 

Thank you. 

Discussion: 

14) Overall it is a good discussion outlining the strengths and limitations of the study 

Thank you. 

15) Please add the disclaimer about the sample group distribution 

We have added this in the limitations. 

16) there are two "most" in the discussion. Please add specific numbers 

We have added the numbers. 

Thanks again and we hope that the above addresses your comments. 

Best regards, 

Stephanie Aboueid, RD, MSc, PhD 

School of Public Health and Health Systems 

University of Waterloo 

Ontario, Canada

---

## [Editor Report · Decision Letter 1]

21 Oct 2021

Latent Classes Associated with the Intention to Use a Symptom Checker for Self-Triage

PONE-D-21-24082R1

Dear Authors,

We’re pleased to inform you that your manuscript has been judged scientifically suitable for publication and will be formally accepted for publication once it meets all outstanding technical requirements.

Kind regards,

Dejan Dragan, PhD

Academic Editor

PLOS ONE

Additional Editor Comments (optional):

The authors have completed a revision of their paper. Already an innovative advanced study has become even better after carefully processed corrections carried out by the authors. Moreover, the answers to reviewers are clear, exact, and transparent. Accordingly, I warmly recommend the acceptance of the paper. The A.E. Dejan Dragan.
---

## [Editor Report · Acceptance letter]

25 Oct 2021

PONE-D-21-24082R1 

Latent Classes Associated with the Intention to Use a Symptom Checker for Self-Triage 

Dear Dr. Aboueid:

I'm pleased to inform you that your manuscript has been deemed suitable for publication in PLOS ONE. Congratulations! Your manuscript is now with our production department. 

Kind regards, 

on behalf of

Dr. Dejan Dragan 

Academic Editor

PLOS ONE